# Hydrophobic Recovery of PDMS Surfaces in Contact with Hydrophilic Entities: Relevance to Biomedical Devices

**DOI:** 10.3390/ma15062313

**Published:** 2022-03-21

**Authors:** Tomoo Tsuzuki, Karine Baassiri, Zahra Mahmoudi, Ayyappasamy Sudalaiyadum Perumal, Kavya Rajendran, Gala Montiel Rubies, Dan V. Nicolau

**Affiliations:** 1Faculty of Engineering and Industrial Science, Industrial Research Institute Swinburne, Swinburne University of Technology, Melbourne, VIC 3122, Australia; colorble@gmail.com; 2Department of Bioengineering, Faculty of Engineering, McGill University, Montreal, QC H3A 0E9, Canada; karine.baassiri@mail.mcgill.ca (K.B.); zahra.mahmoodi@mail.mcgill.ca (Z.M.); ayyappasamy.sudalaiyadumperumal@mcgill.ca (A.S.P.); kavya.rajendran@mail.mcgill.ca (K.R.); gala.montielrubies@mail.mcgill.ca (G.M.R.)

**Keywords:** dimethylpolysiloxane, Atomic Force Microscopy, hydrophobic recovery, biomedical devices, catheters, gas embolism

## Abstract

Polydimethylsiloxane (PDMS), a silicone elastomer, is increasingly being used in health and biomedical fields due to its excellent optical and mechanical properties. Its biocompatibility and resistance to biodegradation led to various applications (e.g., lung on a chip replicating blood flow, medical interventions, and diagnostics). The many advantages of PDMS are, however, partially offset by its inherent hydrophobicity, which makes it unsuitable for applications needing wetting, thus requiring the hydrophilization of its surface by exposure to UV or O_2_ plasma. Yet, the elastomeric state of PDMS translates in a slow, hours to days, process of reducing its surface hydrophilicity—a process denominated as hydrophobic recovery. Using Fourier transform infrared spectroscopy (FTIR) and atomic force microscopy (AFM), the present study details the dynamics of hydrophobic recovery of PDMS, on flat bare surfaces and on surfaces embedded with hydrophilic beads. It was found that a thin, stiff, hydrophilic, silica film formed on top of the PDMS material, following its hydrophilization by UV radiation. The hydrophobic recovery of bare PDMS material is the result of an overlap of various nano-mechanical, and diffusional processes, each with its own dynamics rate, which were analyzed in parallel. The hydrophobic recovery presents a hysteresis, with surface hydrophobicity recovering only partially due to a thin, but resilient top silica layer. The monitoring of hydrophobic recovery of PDMS embedded with hydrophilic beads revealed that this is delayed, and then totally stalled in the few-micrometer vicinity of the embedded hydrophilic beads. This region where the hydrophobic recovery stalls can be used as a good approximation of the depth of the resilient, moderately hydrophilic top layer on the PDMS material. The complex processes of hydrophilization and subsequent hydrophobic recovery impact the design, fabrication, and operation of PDMS materials and devices used for diagnostics and medical procedures. Consequently, especially considering the emergence of new surgical procedures using elastomers, the impact of hydrophobic recovery on the surface of PDMS warrants more comprehensive studies.

## 1. Introduction

Polydimethylsiloxane (PDMS), a silicone elastomer, is a material largely used in many health-related applications, such as in medical interventions (e.g., catheters [1], implants [2]); in medical studies (e.g., atherosclerosis [3], vascularized microfluidics [4], organoids [5], drug response [6]); dentistry (e.g., prostheses [7,8]); in diagnostics (e.g., lab-on-a-chip [9], lateral-flow assays [10]); and in high throughput screening (e.g., microarrays [11]). The extensive use of PDMS in all aspects of biomedical engineering is motivated by its remarkable properties [12]: (i) simple, fast and accurate replication of solid structures, at nm-precision, by the pre-polymer, followed by thermal curing for device consolidation; (ii) maintenance of elastomeric flexibility for very long periods of time; (iii) lack of biotoxicity, if properly processed; (iv) optical transparency; and (v) low autofluorescence, which is critical for fluorescence-based investigations.

The large panoply of advantages associated with PDMS is, however, balanced by few drawbacks, the most important being its hydrophobicity, which is nevertheless essential for the cohesiveness of the polymer. Indeed, in most instances, PDMS surface must be hydrophilized, to make it amenable for interfacing with biological media [13], either aqueous fluids, such as plasma, blood, or buffered solutions containing analytes (DNA, proteins, antibodies, markers, etc.), or cells and tissues. The hydrophilization of PDMS surface consists in the oxidation of the Si-C bonds of the polymer, usually by UV irradiation or oxygen plasma, which leads to the creation of hydrophilic -OH groups on the polymer surface. Initial hydrophilization can be followed by further surface chemistry modification with other functional groups.

While hydrophilization of PDMS surfaces solves most issues related to compatibility with biological media, the very nature of polymer chains in elastomeric state renders the process reversible. Indeed, as PDMS is, at room temperature, well above the temperature of its glassy state, Tg (i.e., approximately −130 °C), the polymer chains are mobile. Consequently, if different regions of PDMS have different chemical compositions upon hydrophilization, the mobility of the polymer chains will induce their cross-diffusion. This process, which involves not only local chemical, but also mechanical changes, is denominated as “hydrophobic recovery”. 

In most instances, the deleterious effects of hydrophobic recovery are mitigated by interfacing the hydrophilized surface of PDPS with aqueous solutions, thus stalling the diffusion of hydrophilic zones in hydrophobic ones, or more rarely by refrigeration of PDMS-made devices at low temperatures, thus decreasing the diffusivity of polymeric chains (which cannot, however, be effectively stopped above Tg). 

Despite extensive studies [14,15,16,17,18,19], there are still unclear aspects of the process of hydrophobic recovery, especially regarding the interplay between the chemical and mechanical parameters of the surface, and the dynamics of the related processes. Additionally, the impact of hydrophobic recovery of PDMS on the functionality of devices and materials using in diagnostics, medical interventions, dentistry, and in vitro studies was not comprehensively explored.

To this end, the present study focuses on the use of Atomic Force Microscopy (AFM) and Fourier Transform Infrared Spectroscopy (FTIR) to monitor the hydrophobic recovery of PDMS at the interface with resilient hydrophilic domains. Results are then discussed in the context of PDMS devices used for catheterization processes and microfluidic devices replicating cardiovascular flow.

## 2. Materials and Methods

*Surface preparation*. Polydimethylsiloxane (PDMS), a silicone elastomer (Sylgard 184, from Dow Corning, Midland, MI, USA) was used to prepare the surfaces. The samples were obtained by mixing the silicone elastomer with the curing agent at a mass ratio of 10:1. After degassing the mixture in a vacuum desiccator for 1 h, all PDMS samples were divided into two categories for analysis: (i) PDMS samples without embedded beads, and (ii) PDMS samples with randomly dispersed beads. 

Two types of beads were used for the second class of surfaces: (i) glass beads with a diameter of 3 to 10 μm (Polyscience Inc., Warrington, PA, USA), and (ii) silica beads with a smaller diameter of 1.5 to 1.9 μm (Spherotech Inc., Lake County, IL, USA). The beads were first mixed with ethanol, after which a small volume of the ethanol-beads mixture was placed on a surface to ensure ethanol evaporation. A mass of 0.5 g of pre-polymer PDMS was subsequently poured over the surface-immobilized beads before being cured in the oven at 65 °C for 3 h. The PDMS with embedded beads was peeled off from the basal surface, then pressed against a glass slide for imaging and analysis with the embedded beads facing up. These surfaces were fabricated in large-enough numbers to allow the analysis at different stages of hydrophilization and subsequent hydrophobic recovery. The procedure of embedding beads on top of PDMS surfaces was presented in detail elsewhere [11].

*Hydrophilization treatment and evolution of hydrophobic recovery*. A house-made sealed cabinet comprising UVC lamps (from Heraeus Amba Lamps, with emission peaks at 254 nm and 185 nm) was used for the treatment of both flat and bead-embedded PDMS. The samples were placed at 2 cm from the radiation source and exposed to UV light for 3 h at room temperature. The UV radiation generated ozone, which in turn reacted with the PDMS surface. Following hydrophilization, both flat and microbead-embedded PDMS surfaces were placed in the same sealed cabinet without irradiation, and analyzed at various times, first every 30 min, then after 6 h, after 24 h, and 48 h, respectively.

*Contact angle measurements*. The surface tension of several biologically relevant fluids, i.e., Fetal Bovine Serum (FBS), Dulbecco’s Modified Eagle Medium (DMEM), DMEM + 10% FBS, whole bovine blood, as well as MiliQ water used as control, was measured using Wilhelmy plate method using a Dynamic Contact Angle Tensiometer (DCAT 11, from Dataphysics Instruments GmbH, Filderstadt, Germany). The surface hydrophobicity of the flat PDMS samples, i.e., without embedded beads, was evaluated by measuring the water contact angle with a Rame-Hard contact angle goniometer (Model 100–115, Rame-Hart, Succasunna, NJ, USA). Static contact angle measurements were performed with the sessile drop technique using 2 μL droplets of Nanopure water (18.2 MΩ·cm^−1^) at room temperature.

*Chemical analysis of the surface*. FTIR spectroscopy in the Attenuated Total Reflection (ATR) mode, performed on a Thermo Nicolet Nexus spectrophotometer (Nicolet Nexus, from Thermo, Waltham, MA, USA) fitted with a SMART SAGA accessory with a Ge crystal (80°), was used to reveal the chemistry of the top layers of the flat PDMS surface, prior, immediately after, and in various periods of time after hydrophilization. An average of 64 scans at a resolution of 2 cm^−1^ formed the FTIR-ATR spectra.

*Microscopy*. An AFM Explorer system (from Thermo Microscopes, Sunnyvale, CA, USA) coupled with the SPMLab Version 5.01 software (from Thermo Microscopes, Sunnyvale, CA, USA), was used for both topography and force measurements, on PDMS surfaces both flat and with embedded beads. The AFM was operated in the contact mode with probes consisting of silicon nitride cantilevers having pyramidal tips and a spring constant of 0.032 N/m. The scanning of the surface produced both topography and lateral force measurements. Additionally, force-distance experiments were performed, resulting in force modulation data. The images were captured through an 8 μm Z-linearized scanner (100 μm ×100 μm) at a temperature of 23 °C and relative humidity of 45%. Background subtraction and brightness and contrast adjustment were performed to analyze the topographical images by using the WSxM V3.0 Beta 9.3 program (from Nanotec España, Elche, Spain). 

The Scanning Electron Microscopy (SEM) images of the PDMS surface impregnated with silica microsphere were obtained using a Quanta FEI450 SEM system (from FEI Company, Hillsboro, OR, USA). The silica side of the PDMS sample surfaces was Pt sputter-coated with Leica EM ACE600 sputter coater (from Leica, Wetzlar, Germany) to a thickness of 4.5 nm. Following Pt coating, the sample was mounted onto sample holders in the SEM stage and topography resolved at high vacuum, imaging voltage of 10 kV.

Figure 1 presents an illustrative topographic image of the unexposed micro-embedded-PDMS surface.

## 3. Results 

### 3.1. Hydrophobic Recovery on Flat PDMS Surfaces

*Contact angle*. The measurement of the surface tension of biologically relevant liquids led to their classification into two groups: (i) highly hydrophilic liquids (i.e., MiliQ water), and DMEM, with surface tensions of 72.75 mN·m^−1^, and 72.89 mN·m^−1^, respectively; and (ii) mildly hydrophilic liquids (i.e., FBS, DMEM + 10% FBS) and bovine whole blood, with surface tensions of 51.65 mN·m^−1^, 52.65 mN·m^−1^, and 53.21 mN·m^−1^, respectively. All previous works regarding the hydrophobic recovery of PDMS report surface hydrophobicity measured via the contact angle of water droplets. Consequently, the mildly hydrophilic liquids (i.e., FBS, DMEM + 10% FBS, and whole blood (as well as DMEM)), were considered only regarding their likely impact on the wetting of PDMS in its hydrophobic and hydrophilized states.

The exposure of flat PDMS surfaces to UV radiation led to a very large drop in the contact angle of water, from 120° to essentially 0° (Figure 2a). Immediately after hydrophilization, the contact angle started to increase, first more rapidly in the first 5–6 h reaching approximately 40°, followed by a more moderate rate of increase, eventually levelling at 45°. 

*Surface roughness*. Similar to the evolution of the contact angle, the exposure of flat PDMS surface to UV radiation led to a considerable increase in the average surface roughness from 10 nm to 40 nm (Figure 2b). Immediately after hydrophilization, the surface roughness decreased dramatically, even below the unexposed value of 10 nm, approximately in the same timespan as the increase of the contact angle (i.e., 5–6 h). Finally, the surface roughness increased at a lower rate towards the end of hydrophobic recovery, reaching approximately 15 nm.

*Force measurements using AFM*. Figure 2c presents the recorded evolution of the AFM force modulation prior to hydrophilization, immediately after, and at different stages of hydrophobic recovery of flat PDMS surfaces. The analysis of force-distance measurements allowed the estimation of Young’s moduli for various stages of hydrophobic recovery (presented in Appendix A). The cantilever amplitude increased on the samples that were scanned immediately after UV exposure, which corresponds to an increased surface hardness, from 1 nA·Å^−1^ to 3 nA·Å^−1^. The calculated Young’s moduli range from 2.4·10^4^ to 1.4·10^5^ depending on indentation depth. 

Finally, Figure 2d presents the evolution of the average lateral force experienced by the AFM tip while scanning the flat PDMS samples, when unexposed, prior to UV hydrophilization, and subsequently at different time intervals. The lateral force presents a steep increase approximately in the same time interval as the abrupt variation of the average roughness (Figure 2b). This was followed by reaching a short maximum at approximately 10 h after hydrophilization, with a continuous decrease to lower values, between untreated, and treated PDMS samples immediately after hydrophilization. 

*FTIR-ATR*. The evolution of the FTIR-ATR spectra for PDMS surface, first unexposed, then immediately after UV treatment, and finally at various periods after, is presented in Figure 3. The considerable increase of the FTIR peaks in the 3050–3800 cm^−1^ region, corresponding to –OH groups [20], immediately after UV treatment, followed by a gradual decrease (especially compared to the spectrum baseline), provides the chemical explanation of the previously observed variation of surface hydrophilicity. A similar, but less apparent evolution occurred for the 725–920 cm^−1^ region, also assignable to –OH groups [20]. The assignment of FTIR peaks in Figure 3 is presented in Appendix A.

### 3.2. Hydrophobic Recovery of PDMS Surfaces Embedded with Hydrophilic Beads

Embedding hydrophilic beads in the hydrophobic PDMS matrix requires the concomitant analysis of topography and lateral force, both provided by AFM scanning. Figure 4 presents the AFM scans for topography (Figure 4a) and lateral force (Figure 4b) of the same 25 µm × 25 µm PDMS surface with embedded beads immediately after hydrophilization. The glass and silica beads appear to aggregate in clusters on the PDMS surfaces, with no apparent rule regarding size distributions, resulting in PDMS domains free of beads, and domains with beads and PDMS between them. The capacity of PDMS to insert between hydrophilic zones is not surprising, as it was demonstrated that it can reproduce the nm-scale details of a mold [21]. Aside from being more elevated (Figure 4a), the areas comprising smaller silica beads or larger glass beads induced a considerably higher lateral force. The hydrophilicity of the beads is represented as high contrast and dark regions, which can be also seen in the lateral force scans of unexposed PDMS surfaces with embedded beads (Figure 5, top left corner). 

The spatial and temporal evolution of surface hydrophilicity, as measured by AFM lateral force is presented in Figure 5. While the hydrophilicity and the stiffness of the beads remain constant throughout the process of hydrophilization and subsequent hydrophobic recovery, as evidenced by the circular areas with an even and constant dark color, the areas surrounding the beads undergo important morphological changes. Initially, the areas away from the hydrophilic beads present a lower lateral force, largely due to the UV-induced stiffness of the top surface, overriding the hydrophilicity-induced increase in lateral force. However, this response is transitory, with the lateral force experienced in areas free of hydrophilic beads returning to lower lateral forces. The probable reason for this evolution is the force-reducing increase in hydrophobicity, which levels back in approximately 4 h. Conversely, the evolution of areas adjacent to the beads is slowed down, with higher lateral forces needed to scan even after long periods of time. Tellingly, the ‘aura’ of lateral force around beads never returned to well-discriminated individual beads, as in unexposed scans (Figure 5, top left), preserving the partially contiguous bead-like lateral force mapping even after 48 h. Another interesting aspect is the clear boundaries between domains, each surrounding a central bead, a characteristic which lasted even after 4 h. Figure 5 also presents a statistical analysis of the lateral force experienced close to the hydrophilic (and rigid) beads, and away from them in an increasingly hydrophobic domain. The surfaces prior to hydrophilization, the lateral forces in the vicinity of hydrophilic beads were essentially identical to the forces away from the beads. Following UV exposure, the recorded lateral forces became considerably higher in regions close to the hydrophilic beads than in PDMS regions away from the beads. A similar evolution of the PDMS with micro-embedded beads, but depicting topography instead of lateral force, is presented in Appendix A.

A quantitative analysis of the lateral force scans is presented as histograms of lateral forces per pixel (Figure 6), and as double lateral force-topography histograms (Figure 7). Notwithstanding the inherent variability from the areas scanned at different times, the histograms of the lateral forces (Figure 6, each panel also presenting the topography of the area used for analysis) suggests that the lateral force has low values immediately after UV exposure (Figure 6, top left), it increases after approximately 4 h, to finally settle in a bimodal distribution of lateral forces experienced by various points on the surface (Figure 6, last row).

The lateral force—topography double histograms (Figure 7) offer a different perspective on the evolution of surface properties. Again, notwithstanding the variations emerging from the areas used for analysis, the double histograms suggest that at the beginning of the hydrophobic recovery the surface exhibits low lateral force values, largely aggregated around higher topographies, consisting of beads and surrounding areas. Subsequently, hydrophobic recovery induces larger values of lateral forces at lower topographic values, due to the increase of the PDMS softness, especially away from the beads. Finally, at longer recovery times, the lateral force spectrum narrows considerably towards lower values, when surface hydrophobicity prevails, with the only high frequency ‘islands’ at higher topographies around and on bead aggregations.

The PDMS surface exhibits low lateral force values around higher topographies consisting of beads and surrounding areas at the onset of hydrophobic recovery. The latter process progresses towards larger lateral forces at lower topographic values due to the increase in PDMS softness, especially at regions distant from the beads. With longer recovery times, the lateral force spectrum continues to narrow towards considerably smaller values, whereby surface hydrophobicity prevails and bead aggregations at higher topographies remain the only high frequency ‘islands’.

A statistical analysis of the spatial distribution of the lateral force, as an indicator of hydrophobic recovery, in the vicinity of the hydrophilic beads is presented in Appendix A.

## 4. Discussion

### 4.1. Hydrophobic Recovery on Flat PDMS Surfaces

*Contact angle*. Contact angle measurement of liquid droplets is arguably the most used method for assessing surface hydrophobicity. Although elaborate mathematical models were derived to correlate the contact angle and the surface tension, and despite the presence of experimental variants (i.e., advancing and receding contact angle measurements), the success of this methodology is largely due to its experimental simplicity. This experimental simplicity comes, however, at a price. Indeed, after ruling out experimental variations, such as the water droplet size (which can be easily standardized), the single value measurement contact angle is a result of an overlap of several experimental and surface physico-chemical parameters. First, the intrinsic surface tension of the substrate has, even in ideal conditions, a complicated relationship with the contact angle [22]. Second, the micro- and possibly nano-topography of the surface modulates the apparent surface hydrophobicity, including the extreme cases of hyper-hydrophobicity and hydrophilicity for materials with similar chemistries [23,24]. Finally, it is often neglected that *flat* surfaces with spatial heterogeneity of hydrophobicity result from micro-, or nano-patches with different chemistry (e.g., self-assembled monolayers [25], single- or double stranded DNA [26], or proteins [27]) modulate the overall hydrophobicity as measured by the contact angle.

This background regarding contact angle, coupled with the changes in surface roughness and chemistry, helps explaining the evolution of the apparent surface hydrophobicity of PDMS in its various states (Figure 2a)—an evolution which was observed in general terms in many studies [28]. The inherent high hydrophobicity of PDMS decreases following O_2_ (or plasma) treatment to nearly total hydrophilicity, with chemical changes on the polymer surface being (almost entirely) responsible for this change. This chemical change and various physico-chemical processes immediately after hydrophilization lead to a gradual increase of the contact angle. However, hydrophobic recovery, as measured by the contact angle, is not complete, since the physico-chemical changes on PDMS surface are not fully reversible.

*Surface roughness*. While surface roughness (Figure 2b) can be measured in various ways, AFM seems to be the natural choice for elastomers [29]. While UV exposure is likely to induce ‘colder’ processes than plasma treatment, the oxidation of the top PDMS layers would be exothermic, and more importantly would result in the variation of the local density of the polymer, thus explaining the increase of the surface roughness. However, immediately after hydrophilization, it is expected that the polymer will relax mechanically, leading to a decrease of the surface roughness, in lockstep with the increase of the contact angle. Finally, slower, physico-chemical and diffusional processes (e.g., movement of silica islands towards the surface) will lead to a moderate increase of the surface roughness.

*Force measurements using AFM*. The versatility of AFM allowed not only the ‘classical’ mapping of surface topography at resolutions not easily available to many optical microscopies, but more importantly, it has also allowed the usage of AFM as a ‘nano laboratory’ [30], including the mechanical and chemical exploration of surfaces, and even nanofabrication. In the context of the present study, one of the most important developments consists in the use of AFM scanning for the measurement of the friction force during scanning [31], particularly on soft surfaces [32], including biological materials [30,33].

The dragging of the AFM tip across the surface in the contact mode will inevitably result in friction. If the AFM tip moves on a direction orthogonal to the axis of the cantilever, i.e., in an ideal case scenario, this friction will result in a torsional force applied to the AFM tip, classically described by Amonton’s Law [34,35,36,37,38] as follows:F_t_ = µ·F_n_ + F_o_(1)
where F_t_ represents the total force, µ is the coefficient of friction, F_n_ is the normal force induced by the AFM cantilever, and F_o_ is the residual force due to the attraction force between the surface and the AFM tip. While this law is very general, the nanoscale environment considerably raises the complexity of these interactions. The impact at the nanoscale is considerably more relevant than the one at the microscale and above. Indeed, while F_n_ will mostly be a representation of the scanned topography, due to the non-orthogonal movement of the tip, it will result in a crosstalk between the two forces. Furthermore, the contact between the AFM tip and the surface is modulated by the local environmental conditions (e.g., surface hydrophobicity). Despite these experimental and theoretical difficulties, the mapping of the friction occurring during scanning, concomitantly with the acquisition of the topography, offers a powerful tool for obtaining better image contrast, and the elimination of artifacts, as well as an assessment of the physics and chemistry of the scanned surface [39,40].

Critically, the scanning of hydrophobic surfaces will result in lower friction than experienced by the AFM tips scanning a hydrophilic surface, especially if hydrophilic tips are used [41] due to the condensation of water on the AFM tip-surface contact [42]. This capability was demonstrated with the ability of AFM scanning to distinguish between patches containing double-stranded more hydrophilic DNA, from patches containing more hydrophobic single-stranded DNA, as well as from the background polymeric surface [26]; and between patches comprising more hydrophilic proteins from the background polymeric material [27]. Additionally, and relevant for the present study, a higher lateral force will be exerted on the AFM tip if it scans a soft layer [43], due to the inherent deeper penetration of the tip into the substrate. This aspect is relevant, as the oxidation of PDMS will result in a stiffer material, and thus a higher Young’s modulus [44], likely due to the increase in the content of silica domains in the polymeric chains. To this end, the correlation between the AFM experienced during scanning can be approximately expressed as follows:F = a·(surface hydrophilicity) + b·(surface stiffness)^−1^(2)
where a and b are empirical constants, and where capillary force can explain the first term, and contact mechanics the second.

Capillary forces will have a minor contribution to the *vertical* force modulation (e.g., ‘snapping’ of the AFM tip on hydrophilic surfaces). Furthermore, with this contribution decreasing during hydrophobic recovery, the major impact on force modulation can be ascribed to the stiffness of the scanned material. From this perspective, it is surprising that the stiffness of the material continues to increase until reaching a maximum at approximately 24 h after hydrophilization. This unexpected evolution could be the result of the ‘consolidation’ of the material supporting the hydrophilic skin, due to the slow crosslinking of the polymer underneath, whose chains were broken during UV exposure.

The evolution of the AFM force modulation (Figure 2c), that is, a gradual increase up to approximately 24 h after hydrophilization, and a decrease afterwards, can be explained by the ‘tug of war’ between the decrease of the oxidized PDMS stiffness, leading to an increase of the force required to scan the sample due to superficial ‘ploughing’, and the increase of surface hydrophobicity leading to lower friction, results in a variation where the maximum is reached after around 48 h after exposure. Importantly, the values of the Young’s modulus are smaller than those expected for PDMS. This observation can be corelated to an earlier study that used AFM indentation on a PDMS surface treated with oxygen plasma [44], reported a significant increasing of the surface hardness (~10 times). Another study [45], which used a slow positron beam analysis, suggests that an 80 nm silica layer formed on the top PDMS surface following UV exposure, laying on thicker (180–360 nm) polymer stratum comprised of lower molecular weight chains. In this context, it is remarkable that the force-distance (F-d) measurements identified a top layer, approximately 80 nm thick (Appendix A), where Young’s modulus was considerably different than that of the material beneath.

It is important to underline that PDMS has inherent properties amenable to blood vessel mimicry from the point of view of mechanical properties [46,47] and diffusivity of O_2_ [48]. For instance, and notwithstanding its variation with the strain, the Young’s modulus of PDMS ranges from 0.05 to 2 MPa [49], and up to 5 MPa [50], compared with 0.04 to 2 MPa for human abdominal aorta sections, and 0.05 to 1.45 MPa for human iliac artery [51], and in special cases up to 5 MPa [52]. The Young’s modulus of venous blood vessels is usually lower (i.e., between 0.05 and 2 MPa) [52]. However, in the context of this study, a major characteristic of blood vessels is their hydrophilicity, which is the result of various proteins being present on the walls, complicated by the presence of various islands with reverse hydrophobicity, e.g., fat deposits [53]. Consequently, the construction of biomimetic, PDMS blood vessels need to take into consideration surface characteristics, as well as bulk properties.

The evolution of the lateral force (Figure 2d) appears to be the result of an overlap of the surface hydrophobicity/hydrophilicity, mechanical properties of the polymer surface, and (to a lesser extent for polymers) surface roughness. After hydrophilization, these three parameters evolve at different rates, and they also have contradictory impact on the apparent lateral force. Consequently, it is expected that the AFM-measured lateral force will present an increase, followed by a gradual decrease, with surface hydrophilicity being the main driving factor.

*FTIR-ATR* was used extensively to study the surface and near-surface chemistry of PDMS with respect to its modulation by polymer molecular weight and copolymer composition [54], the impact of hydrophilization by plasma or ozone treatment [20,55,56], and finally the analysis of the process of hydrophobic recovery [54,55,56]. While the increase of the –OH relevant peaks in the FTIR-ATR spectrum is to be expected when the silicon polymer is exposed to highly energetic, O_2_ gas, of particular relevance is the difference in the evolution of these peaks at different wavelengths. Indeed, the FTIR-ATR penetration depth is inversely proportional to wavenumbers. Using the penetration depth of FTIR-ATR of approximately 1µm for the carbonyl region, i.e., 1900 and 1600 cm^−1^ [54], translates into a penetration depth of approximately 2µm and 0.5 µm, for the –OH regions at 725–920 cm^−1^ and 3050–3800 cm^−1^, respectively. Consequently, and aside from the multitude of FTIR bands in the 725–920 cm^−1^ region, it can be inferred that the larger variation of the –OH bands in the 3050–3800 cm^−1^ region compared with that in the 725–920 cm^−1^ region reflects the larger variation of these chemical groups responsible for surface hydrophilicity at the very top layer of the PDMS surface, as opposed to the immediate ~2 µm thick layer underneath. Indeed, the inset in Figure 3 (larger image in Appendix A) presents the decrease of the *ratio* of the –OH FTIR signals biased towards the top of the surface (approximately 0.5 µm), and those biased more towards the inside the surface (approximately 2 µm), respectively. This evolution indicates a relative decrease of the –OH groups at the top of PDMS surface diffusing inside the film, but it also shows that this process reaches an equilibrium after approximately 6 h.

### 4.2. Hydrophobic Recovery of PDMS Surfaces Embedded with Hydrophilic Beads

According to the experimental observations described above, as well as reported in the literature [19,57,58], the hydrophilization and subsequent hydrophobic recovery of *flat* PDMS surfaces are the result of various overlapping processes (Figure 8, left column):(i)*Chemical processes leading to hydrophilization.* The exposure of flat PDMS material to UV radiation (or O_2_ plasma) translates in the oxidation of the very top layers of the elastomer, thus increasing the SiO_x_ groups embedded in the polymeric matrix. Longer, and/or more intense UV exposure will lead to further breakage and oxidation of the polymeric chains and the formation of silica islands, and eventually a contiguous silica thin ‘skin’.(ii)*Depth limitations of hydrophilization.* The oxidative process cannot progress throughout the thickness of the polymeric film, regardless of the intensity or length of the UV radiation, since PDMS is not transparent to UV light (and cannot be penetrated by plasma-ionised gas). This limitation can be inferred from the more prominent FTIR -OH bands at larger wavenumbers with lower ATR penetration depths than those at smaller wavenumbers with longer penetration depths (Figure 3, first two spectra at the top).(iii)*Mechanical properties of the hydrophilized layer.* The SiO_x_-rich material at the top layer of the exposed PDMS surface presents three important characteristics distinguishing it from the unexposed material. First, the top layer is hydrophilic, which can be inferred from the drastic decrease in contact angle measurements (Figure 2a). Second, the difference in densities between the SiO_x_-rich layer and the submerged native PDMS material results in a wrinkled surface with higher roughness at the nm-scale (Figure 2b). Third, the top layer exhibits more stiffness as shown by the change in Young’s modulus immediately after UV exposure (Figure 2c).(iv) *Different rates of parallel processes following hydrophilization.* Immediately after hydrophilization these three parameters follow a gradual return towards the values prior to UV exposure, each with its own dynamics. This complex process is globally denominated as “hydrophobic recovery”, although it involves more aspects than simply surface hydrophobicity.a.First, the top surface is levelled, driven by the relatively fast mechanical relaxation of the unoxidized, base PDMS elastomer, as evidenced by the initially rapid decrease of the rugosity (Figure 2b). This process also leads to an increase in the Young’s modulus, as well as contributing to an initially more rapid increase of the contact angle (Figure 2a).b.Second, the apparent stiffness increases until approximately 24 h after hydrophilization, (Figure 2c), possibly due to the ‘consolidation’ of the material supporting the hydrophilic skin, due to the slow crosslinking of the polymer underneath, whose chains were broken during UV exposure. c.Third, and concomitantly, a slower process occurs, consisting of the cross-diffusion of the SiO_x_-rich polymeric chains into the native PDMS, and vice versa, resulting in a gradual increase of the Young’s modulus and a slower increase of the contact angle after approximately 10 h (Figure 2c). (v)*Evolution of lateral force during hydrophobic recovery.* The increase of the Young’s modulus and the decrease of hydrophilicity modulate the forces experienced by the AFM tip during scanning, but with different weights and at different rates. Indeed, the *vertical* force modulation (Figure 2c) is impacted more by the increase of Young’s modulus than the *lateral* force (Figure 2d), which in turn is more impacted by surface hydrophobicity than force modulation. The result of this ‘tug of war’ between the increase of Young’s modulus, leading to the increase of the forces experienced by the AFM tip, and the decrease of hydrophilicity, leading to the decrease of these forces, translates to both forces presenting a variation with a maximum (Figure 2c,d). However, due to their different weights, these maxima are located at different times of the hydrophobic recovery (i.e., approximately after 24 h for force modulation) and after 10 h for lateral force.(vi)*End of hydrophobic recovery.* These processes reach a quasi-equilibrium state, but the apparent contact angle does not recover totally back at values prior to hydrophilization. This hysteresis can be explained by the limited depth of diffusion of SiO_x_-rich polymeric chains in the more hydrophobic PDMS base, but also by the fully oxidised silica islands, or even contiguous thin hydrophilic (and stiff) top layer for prolonged, or more intense UV exposure.

To a large extent, these processes also occur for the hydrophilized PDMS in the presence of hydrophilic beads (Figure 8, right column). However, these processes are delayed in the vicinity of regions with resilient hydrophilicity, as evidenced by the larger lateral forces experienced near the beads than away from them (Figure 5).

One of the interesting aspects of the study of hydrophobic recovery in the presence of hydrophilic beads is that it offers the possibility to estimate the thickness of the layer in which the hydrophilicity is resilient. Indeed, while the FTIR can offer only a qualitative estimation of this thickness, the lateral force AFM scan (Figure 5, bottom right) reveals a ‘hydrophilic aura’ around hydrophilic beads of approximately 2–3 µm, depending on the number of beads forming aggregates. The hydrophilic hysteresis also manifests by the contiguous nature of the hydrophilic domains surrounding the bead aggregates, as opposed to the hydrophilic beads being isolated when embedded in PDMS prior to UV exposure (Figure 5, top left).

A remaining question is if different concentration and spatial distribution of the hydrophilic beads will impact differently on the hydrophobic recovery. The statical analysis of the spatial distribution of the lateral force around individual, and aggregates of hydrophilic entities (Appendix A, also studied before [59]) indicated that, indeed, the delay in hydrophobic recovery, measured by the ‘aura’ of higher lateral force, is higher around aggregates of beads than around individual ones. However, this variation is nearly perfectly linear, suggesting that the hydrophobic recovery is simply proportional with the extent of the hydrophilic islands.

### 4.3. Relevance of Hydrophobic Recovery for Biomedical Applications 

Two aspects of hydrophobic recovery are relevant for the design, fabrication and operation of PDMS-made materials and devices used for diagnostics, medical procedures: (i) surface chemistry and hydrophobicity; and (ii) surface mechanical properties, as well as their dynamics. 

*Surface chemistry and hydrophobicity*. The present study shows that hydrophobic recovery is stalled in the near vicinity of the interface between the hydrophilized surface of PDMS and a hydrophilic domain, such as hydrophilic glass (which is the driving force of sealing PDMS on glass). Similar stalling of hydrophobic recovery occurs if PDMS structures are preserved in aqueous media [60]. However, even if the hydrophilized PDMS is not protected, the present study shows that, if the oxidation process is strong enough to create a contiguous, thick-enough silica layer, the hydrophobic recovery presents a distinct hysteresis, leading to a final moderate surface hydrophilicity. To conclude, strictly from the chemical point of view, and if precautions are taken, the hydrophobic recovery can be stalled or mitigated for *flat* PDMS surfaces not exposed to substantial mechanical stresses, such as larger areas in microfluidic and organoid-on-a-chip devices.

It is important to qualify these observations from the perspective of the interplay between the surface tension of various biologically-relevant liquids and PDMS material. For instance, there are many instances where the hydrophobicity of PDMS is *desired* (e.g., for avoiding the contact with hydrophilic liquids) such as water containing soluble medication. However, there are other instances when PDMS hydrophobicity contradicts, at least partially, the avoidance of the contact with flowing fluids if these fluids present a moderate hydrophilicity, such as blood (as measured in the present study). Moreover, blood contains a multitude of heterogenous elements (e.g., red blood cells, fat deposits) which could lead to deleterious effects if adsorbed on the hydrophobic PDMS walls. Consequently, the hydrophobicity, or hydrophilicity of the PDMS conduits carrying biologically-relevant fluids must be carefully tailored along the actual surface tension of these fluids.

*Surface mechanical properties*. A much more serious, but often neglected impact on the design, fabrication, and operation of PDMS-made structures is related to the deformations induced by the intrinsic mechanical processes that occur during hydrophilization and subsequent hydrophobic recovery, or by the mechanical forces applied during the operation of PDMS-devices and materials. It was shown, in this study and in other various reports [16,19,58], that the hydrophilization of PDMS leads to the formation of a thin (below 100 nm) hydrophilic silica layer, on top of a thicker polymeric layer with UV-altered chains, which in turn rests on top of the unaffected hydrophobic PDMS. All of these layers have different characteristics, among the most relevant being the different densities and Young’s moduli. 

The first impact of the vertical heterogeneity of the mechanical physico-chemical and mechanical properties will manifest immediately after hydrophilization and during hydrophobic recovery, due to *internal* reorganization forces. For instance, the actual geometry of the microfluidic structures will depart from their design, especially if nm-level precision is needed for the operation of the respective devices [61]. A more dramatic impact will occur if the hydrophilized PDMS-made materials and devices are exposed to *external* forces. For instance, PDMS-made tubing and catheters, which in many instances must be hydrophilized to allow the facile interfacing with biological fluids, are exposed to external forces due to the very nature of their use in medical procedures. These forces will very likely lead to the breakage of the stiff, but very thin, hydrophilic silica ‘skin’, thus exposing the hydrophobic core of PDMS material. While due to the elastomeric nature of PDMS, these effects will be very localized, the hydrophobic islands can cascade in important deleterious effects. In this context, and as mentioned earlier, it was shown [53] that hydrophobic patches (e.g., fat, thrombi) can constitute important nucleation points for the generation of gas emboli or even growth of deposits leading to the obstruction of blood flow and restriction of the vascular transport. Consequently, similar occurrences involving PDMS tubing, in particular if inserted in, or placed in contact with blood vessels [62] could lead to deleterious medical outcomes, thus warranting additional comprehensive studies. 

## 5. Conclusions

The present study used FTIR-ATR and more extensively AFM techniques to reveal the dynamics of hydrophobic recovery of PDMS, as a flat bare surface and embedded with hydrophilic beads. It was found that the UV radiation-based hydrophilization induced the formation of a thin, stiff, hydrophilic silica film on top of the PDMS material. The hydrophobic recovery of bare PDMS appears to be an overlap of various nano-mechanical, and diffusional processes, each with its own rate dynamics, with a clear hysteresis, with surface hydrophobicity recovering only partially due to a thin, but resilient top silica layer. The monitoring of micro-embedded-PDMS hydrophobic recovery on the other hand, revealed that this is delayed and totally stalled in the very few micrometers vicinity of the hydrophilic domains, which can be used as a good approximation of the depth of the resilient moderately hydrophilic top layer. Since this work presents the effects of hydrophilization and subsequently hydrophobic recovery of PDMS, generally used in biomedical and health-related applications, the impact of these results can be extended to deepen the study of vascular biology and chronical diseases associated with hydrophobic islands.

## Figures and Tables

**Figure 1 materials-15-02313-f001:**
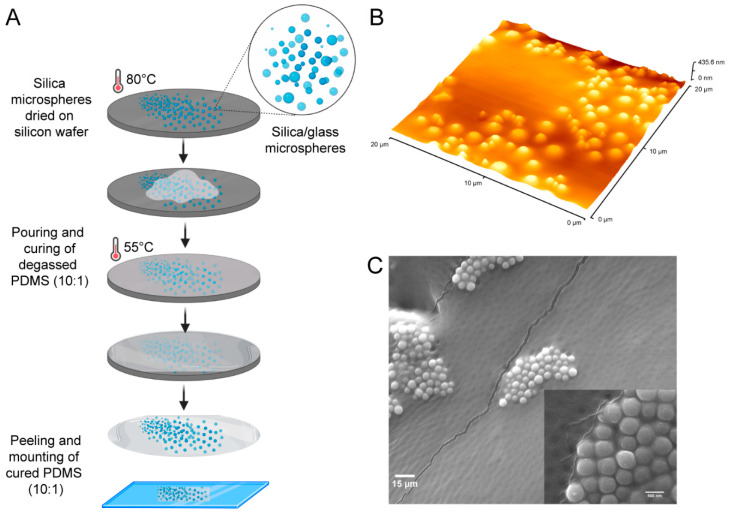
(**A**) Process of the fabrication of PDMS surfaces embedded with hydrophilic particles. (**B**) Topography of PDMS surface with embedded hydrophilic particles. (**C**) SEM image of the surface of PDMS with aggregates of beads.

**Figure 2 materials-15-02313-f002:**
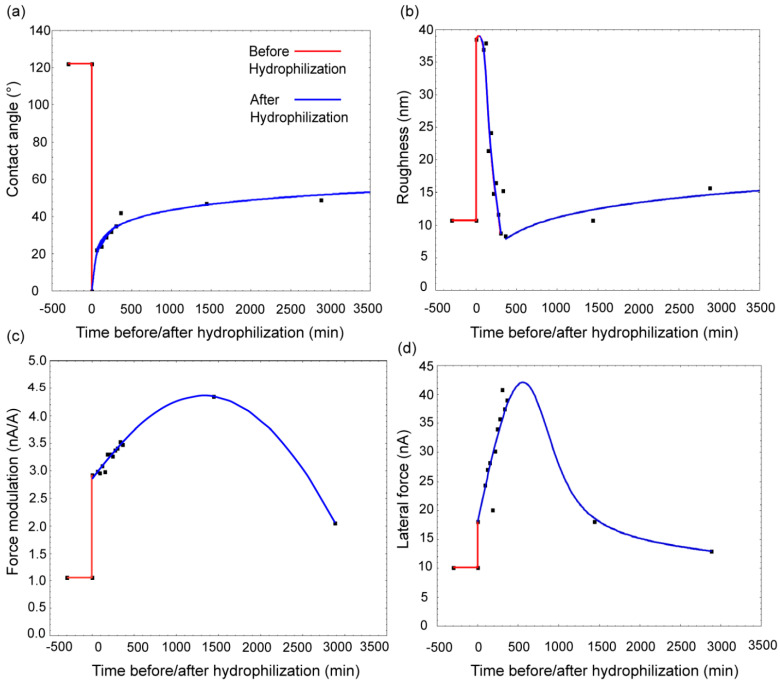
Evolution of physico-chemical properties of *flat* PDMS surfaces, prior to hydrophilization (red) and subsequently, i.e., during hydrophobic recovery (blue). (**a**) contact angle; (**b**) roughness; (**c**) force modulation; and (**d**) lateral force.

**Figure 3 materials-15-02313-f003:**
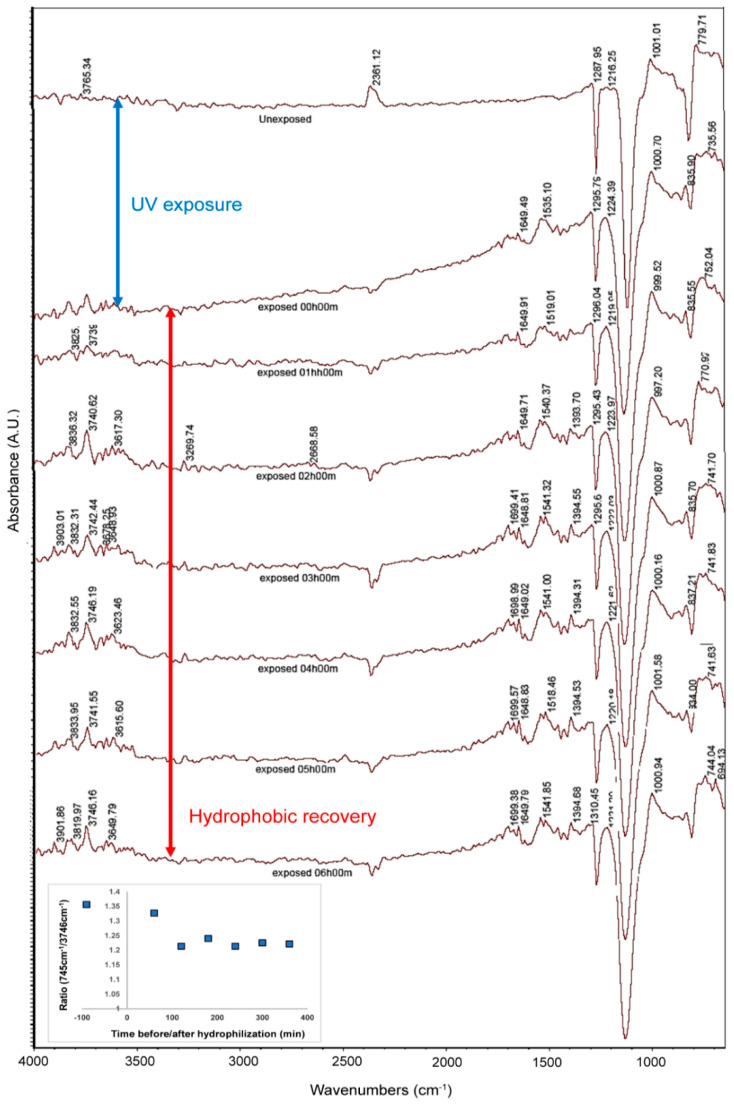
Evolution of the FTIR-ATR spectra of flat PDMS surfaces, before and immediately after hydrophilization, and at different times after.

**Figure 4 materials-15-02313-f004:**
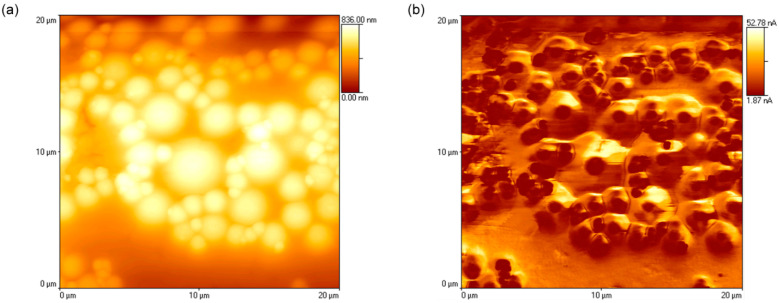
AFM scans of the topography (**a**) and lateral force (**b**) of the hydrophilized PDMS immediately after UV radiation.

**Figure 5 materials-15-02313-f005:**
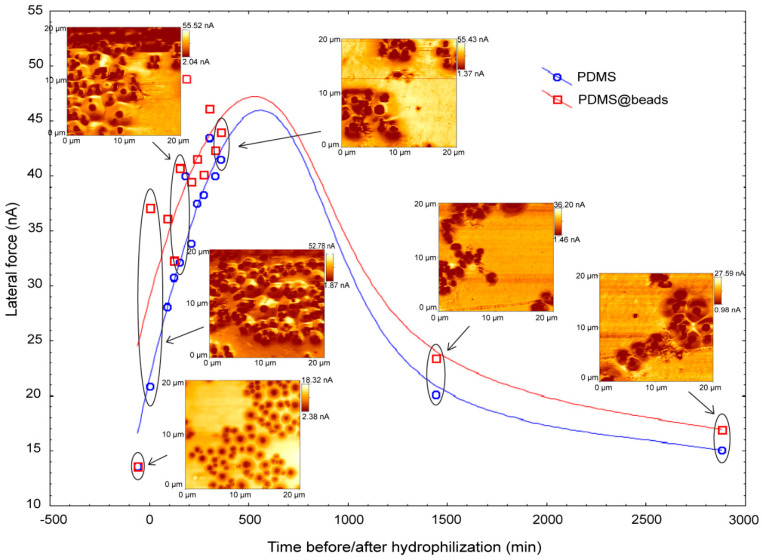
Variation of the lateral force for PDMS surface away (blue) and in the vicinity (red) of hydrophilic entities. The associated later force scans are indicated at the respective positions in time during hydrophobic recovery.

**Figure 6 materials-15-02313-f006:**
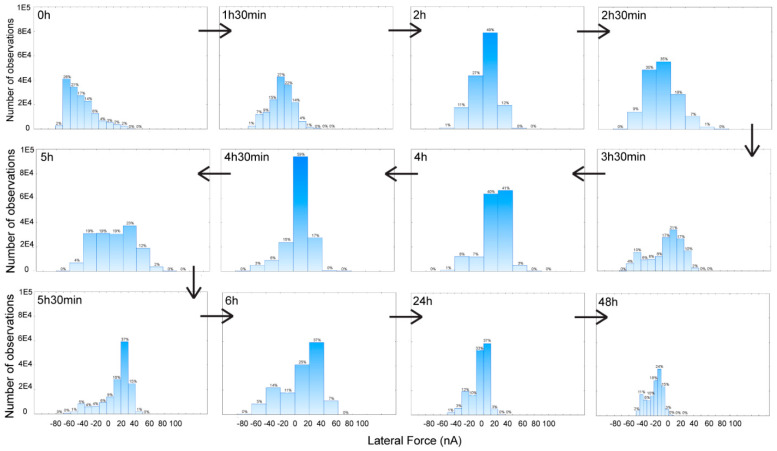
Sequence of histograms of the lateral force readings at various stages of hydrophobic recovery.

**Figure 7 materials-15-02313-f007:**
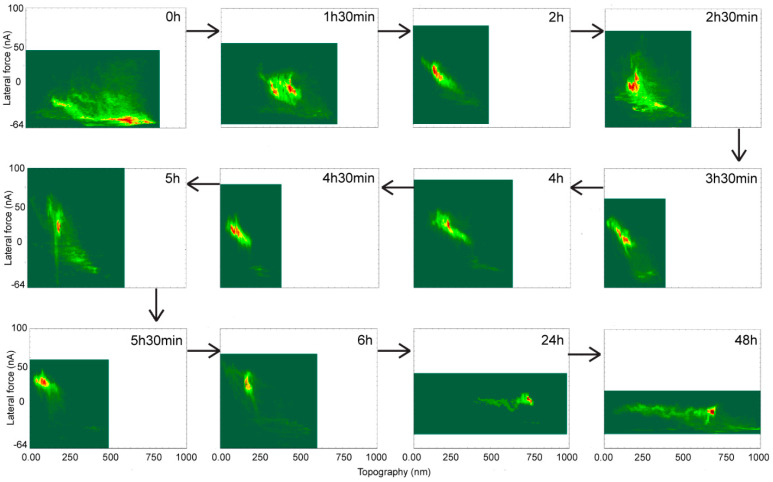
Sequence of double histograms of topography-lateral force readings at various stages of hydrophobic recovery.

**Figure 8 materials-15-02313-f008:**
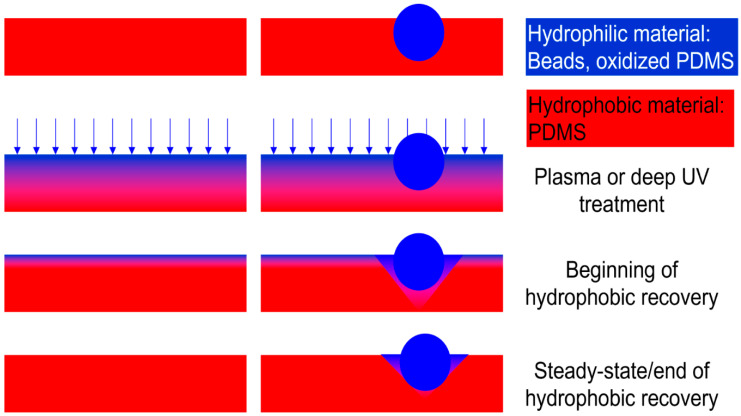
Mechanism of hydrophobic recovery for bare PDMS and PDMS embedded with hydrophilic beads.

## Data Availability

All data is comprised in the published material.

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
