# Peer review of "Hydrophobic Recovery of PDMS Surfaces in Contact with Hydrophilic Entities: Relevance to Biomedical Devices"

_materials, 2022, doi:10.3390/ma15062313_

Round 1

Reviewer 1 Report

The authors present a study monitoring the recovery of PDMS devices using FTIR and AFM using different surfaces embedded with hydrophilic beads. The mechanism of the recovery is proposed and discussed.The paper has studied the dynamics of hydrophobic recovery in PDMS surfaces in detail. The results presented in the work are in depth and the authors have backed their claims with experimental results. The authors should address the following concerns before publication.

  1. The authors present a lot of information about AFM, however I think this is a lot of basic information before coming to the actual experiments in the paper. It reads like an introduction rather than the results and discussion section of the paper. The authors can condense the first two paragraphs in the Force Measurements using AFM paragraph
  2. On Figure 2, the legends for the red and blue are missing. Also, what the figures a, b c, d denotes are missing in the captions. Similarly, in the paragraph on surface roughness, there needs to be space before units such as nm are mentioned. This needs to be uniform throughout the paper.
  3. Can the authors show SEM images of how the beads embedded inside the PDMS layer look?
  4. Can the authors show if ordering the distribution of beads or changing the concentration of the beads impacts the recovery?
  5. Figure 6, the histogram legends are not visible. Can the authors just show certain data points or plot staggered plot as right now the plots too crowded and even the scale bar is not visible.
  6. The authors can combine the figures to convey their conclusions more effectively. Right now, the manuscript has 9 figures many of which are just single figures. Combining them will make it much easier for the readers to understand the paper.
  7. In the mechanism, can the authors make the conclusions more concise. Right now, with 9 points it is difficult to understand what the exact mechanism is.
  8. Overall the paper can be much more concise to convey the conclusions better.
  9. The authors speak about relevance of biomedical applications but no such data is presented in the paper. Can the authors do a study on PDMS with some biologically relevant synthetic fluids like cell culture medium (DMEM, DMEM+FBS etc.) to support their statements.

Author Response

Reviewer: The authors present a study monitoring the recovery of PDMS devices using FTIR and AFM using different surfaces embedded with hydrophilic beads. The mechanism of the recovery is proposed and discussed. The paper has studied the dynamics of hydrophobic recovery in PDMS surfaces in detail. The results presented in the work are in depth and the authors have backed their claims with experimental results.

Authors’ reply. We thank the reviewer for consideration and for suggestions, all being addressed, as follows.

Reviewer: The authors should address the following concerns before publication.

  1. The authors present a lot of information about AFM, however I think this is a lot of basic information before coming to the actual experiments in the paper. It reads like an introduction rather than the results and discussion section of the paper. The authors can condense the first two paragraphs in the Force Measurements using AFM paragraph.

Authors’ reply. We followed this suggestion, and that of the 2nd Reviewer, and decided to split our contribution in separate sections for Results, and Discussion, respectively. Consequently, the results are presenting first, following by an analysis of these results, item by item.

Reviewer:

  1. On Figure 2, the legends for the red and blue are missing. Also, what the figures a, b c, d denotes are missing in the captions.

Authors’ reply. We changed the Figure accordingly.

  1. Reviewer: Similarly, in the paragraph on surface roughness, there needs to be space before units such as nm are mentioned. This needs to be uniform throughout the paper.

Authors’ reply. Throughout the manuscript the style “number”-space-“unit” was implemented.

  1. Can the authors show SEM images of how the beads embedded inside the PDMS layer look?

Authors’ reply. SEM images have been provided in the new, composite Figure 1.

  1. Can the authors show if ordering the distribution of beads or changing the concentration of the beads impacts the recovery?

Authors’ reply. A statistical analysis of the spatial distribution of the lateral force, as a measure of the hydrophilicity, was performed. It was found that the extent of the delay of hydrophobic recovery is in a near perfect linear relationship with the extent of the area of hydrophilic entities.

  1. Figure 6, the histogram legends are not visible. Can the authors just show certain data points or plot staggered plot as right now the plots too crowded and even the scale bar is not visible.

Authors’ reply. Figure 6 was changed considerably. The legends were enlarged and the AFM scans were moved to Supplementary Information to allow an easier inspection of the Figure.

  1. The authors can combine the figures to convey their conclusions more effectively. Right now, the manuscript has 9 figures many of which are just single figures. Combining them will make it much easier for the readers to understand the paper.

Authors’ reply. In the original manuscript only two Figures were ‘single’, all the rest were composite. Additionally, the Reviewer asked for additional information. This being said, we do acknowledge that 9 figures is a number larger than in most published scientific articles. To solve this problem, we (i) made a composite Figure 1, comprising the previous Figure 1 plus additional material required by the Reviewer; and (ii) made a composite Figure from Figure 5 and previous Figure 8, reaching a total number of 8 Figures.

  1. In the mechanism, can the authors make the conclusions more concise. Right now, with 9 points it is difficult to understand what the exact mechanism is.

Authors’ reply. With hydrophobic recovery being an overlap of several, and competing processes, each with different rates, we tried our best to be precise in our proposed mechanism. This being said, we do understand Reviewer’s comment and we clarified what each headline is about. We also collapsed the proposed mechanism in six major modules.

  1. Overall the paper can be much more concise to convey the conclusions better.

Authors’ reply. To achieve the desideratum, we re-organised the whole manuscript in Results and Discussion. Also, the Conclusions, and the Abstract, were made more clear with regard to the findings and the conclusions.

  1. The authors speak about relevance of biomedical applications but no such data is presented in the paper. Can the authors do a study on PDMS with some biologically relevant synthetic fluids like cell culture medium (DMEM, DMEM+FBS etc.) to support their statements.

Authors’ reply. The surface tension o the fluids suggested by the Reviewer, as well as blood, were measured and the values reported. In the Discussion section these values were put in context in relation with the hydrophobicity/hydrophilicity of the actual PDMS-made devices and conduits for the respective fluids.

Reviewer 2 Report

The following suggested must be addressed before reconsideration:

  1. A schematic representation of the modification procedure would be helpful for the readers.
  2. At the end of 'introduction' section, the novelty of the manuscript in comparison with previous published works must be highlighted.
  3. I recommend the split of 'Results and discussions' into two different sections. In 'Results' the authors should present the data from analysis methods that support and prove the surface modification. In 'Discussion' section the authors should comment the obtained results in comparison with previous reported works - what are the differences between the reported results and other hydrophobic methods, what are the advances and disadvantages of present work.

Author Response

Reviewer 2

The following suggested must be addressed before reconsideration:

Authors’ reply. We thank the reviewer for consideration and for suggestions, all being addressed, as follows.

  1. A schematic representation of the modification procedure would be helpful for the readers.

Authors’ reply. Figure 1 comprises now the suggested scheme.

  1. At the end of 'introduction' section, the novelty of the manuscript in comparison with previous published works must be highlighted.

Authors’ reply. The last paragraph, as well as the Abstract, and the Conclusion, was changed along this suggestion.

  1. I recommend the split of 'Results and discussions' into two different sections. In 'Results' the authors should present the data from analysis methods that support and prove the surface modification. In 'Discussion' section the authors should comment the obtained results in comparison with previous reported works - what are the differences between the reported results and other hydrophobic methods, what are the advances and disadvantages of present work.

Authors’ reply. The authors thank the Reviewer for this suggestion, which we followed. We split our contribution in separate sections for Results, and Discussion, respectively. Consequently, the results are presenting first, following by an analysis of these results, item by item. We also highlighted the additional knowledge gained following this study.

Reviewer 3 Report

Tsuzuki et al have presented an interesting study on the hydrophobic recovery of PDMS surfaces. The sound design of experiments consisting of the AFM and FTIR-ATR demonstrated the hydrophobic recovery of PDMS embedded with hydrophilic beads. The results depicted the complex processes of hydrophilization and subsequent hydrophobic recovery. Overall the manuscript is well written and the results are discussed properly. Thus I recommend that the manuscript can be published in its current form.

Author Response

Tsuzuki et al have presented an interesting study on the hydrophobic recovery of PDMS surfaces. The sound design of experiments consisting of the AFM and FTIR-ATR demonstrated the hydrophobic recovery of PDMS embedded with hydrophilic beads. The results depicted the complex processes of hydrophilization and subsequent hydrophobic recovery. Overall the manuscript is well written and the results are discussed properly. Thus I recommend that the manuscript can be published in its current form.

Authors’ reply. The authors thank the Reviewer for consideration. However, following the opinions and suggestions of other Reviewers, the manuscript had to be changed.

Reviewer 4 Report

Dear authors. Congratulations to your manusscript.

Review:

The manuscript describes investigations on PDMS surfaces hydrophilized by UV radiation.

PDMS shows special properties such as thermal stability, softness, biocompatibility and biostability. These properties make PDMS a very suitable material in biomaterial research. A disadvantage of PDMS is that it cannot be easily hydrophilized. The high flexibility of the PDMS network causes the so-called hydrophobic recovery. The manuscript describes how the PDMS surfaces were hydrophilized by means of UV radiation and in which time periods the hydrophobic recovery occurs. By using hydrophilic SiOx particles, it could be shown that in the vicinity of the silica particles, hydrophilic recovery can be suppressed or even eliminated. This approach represents a promising way to generate a certain hydrophilic on PDMS surfaces.

The surfaces were investigated using surface sensitive methods such as contact angle measurement, AFM and IR-ATR. Through intensive AFM studies, the dynamics of the hydrophobic recoveries were investigated and a detailed mechanism was presented.

The investigations are characterised by a consistently high scientific level and are described in great detail and comprehensibly in clear language in the manuscript.

The scientific literature has been considered to a sufficient extent and in appropriate depth. The research results have always been carefully discussed in the light of the relevant literature. All graphs have been discussed in detail and in a comprehensible manner.

Especially successfully is the attempt to present a detailed mechanism of hydrophobic recoveries with the different temporal events.

Author Response

Reviewer 4

Dear authors. Congratulations to your manuscript.

Review:

  1. The manuscript describes investigations on PDMS surfaces hydrophilized by UV radiation.
  2. PDMS shows special properties such as thermal stability, softness, biocompatibility and biostability. These properties make PDMS a very suitable material in biomaterial research. A disadvantage of PDMS is that it cannot be easily hydrophilized. The high flexibility of the PDMS network causes the so-called hydrophobic recovery. The manuscript describes how the PDMS surfaces were hydrophilized by means of UV radiation and in which time periods the hydrophobic recovery occurs. By using hydrophilic SiOx particles, it could be shown that in the vicinity of the silica particles, hydrophilic recovery can be suppressed or even eliminated. This approach represents a promising way to generate a certain hydrophilic on PDMS surfaces.
  3. The surfaces were investigated using surface sensitive methods such as contact angle measurement, AFM and IR-ATR. Through intensive AFM studies, the dynamics of the hydrophobic recoveries were investigated and a detailed mechanism was presented.
  4. The investigations are characterised by a consistently high scientific level and are described in great detail and comprehensibly in clear language in the manuscript.
  5. The scientific literature has been considered to a sufficient extent and in appropriate depth. The research results have always been carefully discussed in the light of the relevant literature. All graphs have been discussed in detail and in a comprehensible manner.
  6. Especially successfully is the attempt to present a detailed mechanism of hydrophobic recoveries with the different temporal events.

Authors’ reply. The authors thank the Reviewer for consideration. However, following the opinions and suggestions of other Reviewers, the manuscript had to be changed.

Round 2

Reviewer 2 Report

The manuscript can be published in current form.